# Gradient Estimation Using
# Stochastic Computation Graphs

**John Schulman**[1,2]
joschu@eecs.berkeley.edu

**Nicolas Heess**[1]
heess@google.com

**Theophane Weber**[1]
theophane@google.com

**Pieter Abbeel**[2]
pabbeel@eecs.berkeley.edu

[1] Google DeepMind      [2] University of California, Berkeley, EECS Department

## Abstract

In a variety of problems originating in supervised, unsupervised, and reinforcement learning, the loss function is defined by an expectation over a collection of random variables, which might be part of a probabilistic model or the external world. Estimating the gradient of this loss function, using samples, lies at the core of gradient-based learning algorithms for these problems. We introduce the formalism of *stochastic computation graphs*—directed acyclic graphs that include both deterministic functions and conditional probability distributions—and describe how to easily and automatically derive an unbiased estimator of the loss function's gradient. The resulting algorithm for computing the gradient estimator is a simple modification of the standard backpropagation algorithm. The generic scheme we propose unifies estimators derived in variety of prior work, along with variance-reduction techniques therein. It could assist researchers in developing intricate models involving a combination of stochastic and deterministic operations, enabling, for example, attention, memory, and control actions.

## 1 Introduction

The great success of neural networks is due in part to the simplicity of the backpropagation algorithm, which allows one to efficiently compute the gradient of any loss function defined as a composition of differentiable functions. This simplicity has allowed researchers to search in the space of architectures for those that are both highly expressive and conducive to optimization; yielding, for example, convolutional neural networks in vision [12] and LSTMs for sequence data [9]. However, the backpropagation algorithm is only sufficient when the loss function is a deterministic, differentiable function of the parameter vector.

A rich class of problems arising throughout machine learning requires optimizing loss functions that involve an expectation over random variables. Two broad categories of these problems are (1) likelihood maximization in probabilistic models with latent variables [17, 18], and (2) policy gradients in reinforcement learning [5, 23, 26]. Combining ideas from from those two perennial topics, recent models of attention [15] and memory [29] have used networks that involve a combination of stochastic and deterministic operations.

In most of these problems, from probabilistic modeling to reinforcement learning, the loss functions and their gradients are intractable, as they involve either a sum over an exponential number of latent variable configurations, or high-dimensional integrals that have no analytic solution. Prior work (see Section 6) has provided problem-specific derivations of Monte-Carlo gradient estimators, however, to our knowledge, no previous work addresses the general case.

Appendix C recalls several classic and recent techniques in variational inference [14, 10, 21] and reinforcement learning [23, 25, 15], where the loss functions can be straightforwardly described using

the formalism of stochastic computation graphs that we introduce. For these examples, the variance-reduced gradient estimators derived in prior work are special cases of the results in Sections 3 and 4.

The contributions of this work are as follows:

- We introduce a formalism of stochastic computation graphs, and in this general setting, we derive unbiased estimators for the gradient of the expected loss.
- We show how this estimator can be computed as the gradient of a certain differentiable function (which we call the *surrogate loss*), hence, it can be computed efficiently using the backpropagation algorithm. This observation enables a practitioner to write an efficient implementation using automatic differentiation software.
- We describe variance reduction techniques that can be applied to the setting of stochastic computation graphs, generalizing prior work from reinforcement learning and variational inference.
- We briefly describe how to generalize some other optimization techniques to this setting: majorization-minimization algorithms, by constructing an expression that bounds the loss function; and quasi-Newton / Hessian-free methods [13], by computing estimates of Hessian-vector products.

The main practical result of this article is that to compute the gradient estimator, one just needs to make a simple modification to the backpropagation algorithm, where extra gradient signals are introduced at the stochastic nodes. Equivalently, the resulting algorithm is *just* the backpropagation algorithm, applied to the surrogate loss function, which has extra terms introduced at the stochastic nodes. The modified backpropagation algorithm is presented in Section 5.

## 2 Preliminaries

### 2.1 Gradient Estimators for a Single Random Variable

This section will discuss computing the gradient of an expectation taken over a single random variable—the estimators described here will be the building blocks for more complex cases with multiple variables. Suppose that $x$ is a random variable, $f$ is a function (say, the cost), and we are interested in computing $\frac{\partial}{\partial \theta} \mathbb{E}_x [f(x)]$. There are a few different ways that the process for generating $x$ could be parameterized in terms of $\theta$, which lead to different gradient estimators.

- We might be given a parameterized probability distribution $x \sim p(\cdot; \theta)$. In this case, we can use the *score function* (SF) estimator [3]:

$$\frac{\partial}{\partial \theta} \mathbb{E}_x [f(x)] = \mathbb{E}_x \left[ f(x) \frac{\partial}{\partial \theta} \log p(x; \theta) \right]. \tag{1}$$

This classic equation is derived as follows:

$$\frac{\partial}{\partial \theta} \mathbb{E}_x [f(x)] = \frac{\partial}{\partial \theta} \int dx \, p(x; \theta) f(x) = \int dx \, \frac{\partial}{\partial \theta} p(x; \theta) f(x)$$

$$= \int dx \, p(x; \theta) \frac{\partial}{\partial \theta} \log p(x; \theta) f(x) = \mathbb{E}_x \left[ f(x) \frac{\partial}{\partial \theta} \log p(x; \theta) \right] \tag{2}$$

This equation is valid if and only if $p(x; \theta)$ is a continuous function of $\theta$; however, it does not need to be a continuous function of $x$ [4].

- $x$ may be a deterministic, differentiable function of $\theta$ and another random variable $z$, i.e., we can write $x(z, \theta)$. Then, we can use the *pathwise derivative* (PD) estimator, defined as follows.

$$\frac{\partial}{\partial \theta} \mathbb{E}_z [f(x(z, \theta))] = \mathbb{E}_z \left[ \frac{\partial}{\partial \theta} f(x(z, \theta)) \right]. \tag{3}$$

This equation, which merely swaps the derivative and expectation, is valid if and only if $f(x(z, \theta))$ is a continuous function of $\theta$ for all $z$ [4]. [1] That is not true if, for example, $f$ is a step function.

- Finally $\theta$ might appear both in the probability distribution and inside the expectation, e.g., in $\frac{\partial}{\partial \theta} \mathbb{E}_{z \sim p(\cdot; \, \theta)} [f(x(z,\theta))]$. Then the gradient estimator has two terms:

$$\frac{\partial}{\partial \theta} \mathbb{E}_{z \sim p(\cdot; \, \theta)} [f(x(z,\theta))] = \mathbb{E}_{z \sim p(\cdot; \, \theta)} \left[ \frac{\partial}{\partial \theta} f(x(z,\theta)) + \left( \frac{\partial}{\partial \theta} \log p(z; \theta) \right) f(x(z,\theta)) \right]. \quad (4)$$

This formula can be derived by writing the expectation as an integral and differentiating, as in Equation (2).

In some cases, it is possible to *reparameterize* a probabilistic model—moving $\theta$ from the distribution to inside the expectation or vice versa. See [3] for a general discussion, and see [10, 21] for a recent application of this idea to variational inference.

The SF and PD estimators are applicable in different scenarios and have different properties.

1. SF is valid under more permissive mathematical conditions than PD. SF can be used if $f$ is discontinuous, or if $x$ is a discrete random variable.

2. SF only requires sample values $f(x)$, whereas PD requires the derivatives $f'(x)$. In the context of control (reinforcement learning), SF can be used to obtain unbiased policy gradient estimators in the "model-free" setting where we have no model of the dynamics, we only have access to sample trajectories.

3. SF tends to have higher variance than PD, when both estimators are applicable (see for instance [3, 21]). The variance of SF increases (often linearly) with the dimensionality of the sampled variables. Hence, PD is usually preferable when $x$ is high-dimensional. On the other hand, PD has high variance if the function $f$ is rough, which occurs in many time-series problems due to an "exploding gradient problem" / "butterfly effect".

4. PD allows for a deterministic limit, SF does not. This idea is exploited by the deterministic policy gradient algorithm [22].

**Nomenclature.**    The methods of estimating gradients of expectations have been independently proposed in several different fields, which use differing terminology. What we call the *score function* estimator (via [3]) is alternatively called the *likelihood ratio* estimator [5] and REINFORCE [26]. We chose this term because the score function is a well-known object in statistics. What we call the *pathwise derivative* estimator (from the mathematical finance literature [4] and reinforcement learning [16]) is alternatively called *infinitesimal perturbation analysis* and *stochastic backpropagation* [21]. We chose this term because pathwise derivative is evocative of propagating a derivative through a sample path.

## 2.2   Stochastic Computation Graphs

The results of this article will apply to stochastic computation graphs, which are defined as follows:

> **Definition 1** (Stochastic Computation Graph). *A directed, acyclic graph, with three types of nodes:*
>
> 1. *Input nodes, which are set externally, including the parameters we differentiate with respect to.*
>
> 2. *Deterministic nodes, which are functions of their parents.*
>
> 3. *Stochastic nodes, which are distributed conditionally on their parents.*
>
> *Each parent $v$ of a non-input node $w$ is connected to it by a directed edge $(v, w)$.*

In the subsequent diagrams of this article, we will use circles to denote stochastic nodes and squares to denote deterministic nodes, as illustrated below. The structure of the graph fully specifies what estimator we will use: SF, PD, or a combination thereof. This graphical notation is shown below, along with the single-variable estimators from Section 2.1.

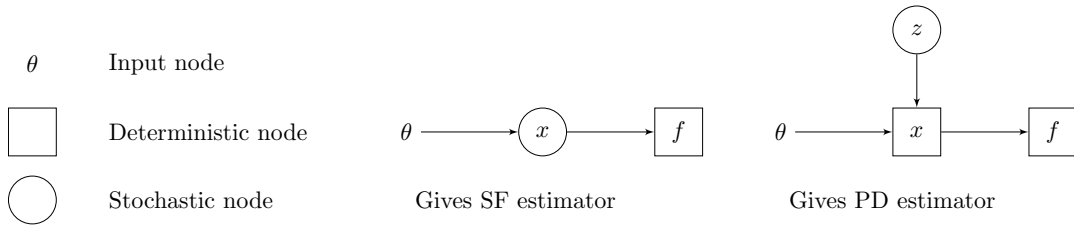

| | |
|---|---|
| $\theta$ | Input node |
| [square] | Deterministic node |
| (circle) | Stochastic node |

Gives SF estimator     Gives PD estimator

## 2.3 Simple Examples

Several simple examples that illustrate the stochastic computation graph formalism are shown below. The gradient estimators can be described by writing the expectations as integrals and differentiating, as with the simpler estimators from Section 2.1. However, they are also implied by the general results that we will present in Section 3.

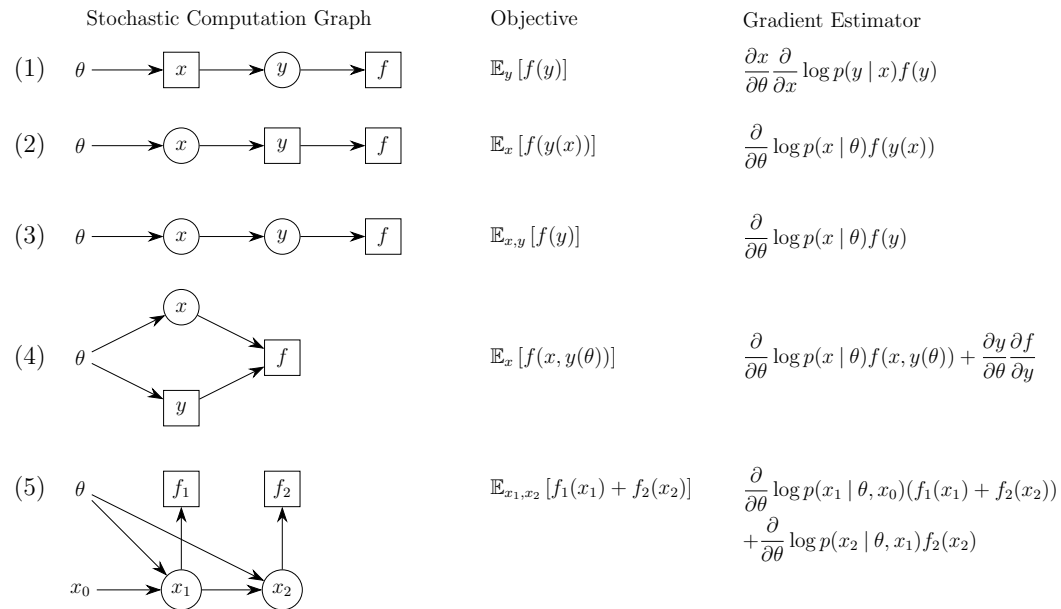

Stochastic Computation Graph      Objective      Gradient Estimator

(1) $\theta \longrightarrow x \longrightarrow y \longrightarrow f$  $\mathbb{E}_y\left[f(y)\right]$  $\dfrac{\partial x}{\partial \theta}\dfrac{\partial}{\partial x}\log p(y\mid x)f(y)$

(2) $\theta \longrightarrow x \longrightarrow y \longrightarrow f$  $\mathbb{E}_x\left[f(y(x))\right]$  $\dfrac{\partial}{\partial \theta}\log p(x\mid\theta)f(y(x))$

(3) $\theta \longrightarrow x \longrightarrow y \longrightarrow f$  $\mathbb{E}_{x,y}\left[f(y)\right]$  $\dfrac{\partial}{\partial \theta}\log p(x\mid\theta)f(y)$

(4) $\theta \longrightarrow x, y \longrightarrow f$  $\mathbb{E}_x\left[f(x,y(\theta))\right]$  $\dfrac{\partial}{\partial \theta}\log p(x\mid\theta)f(x,y(\theta))+\dfrac{\partial y}{\partial \theta}\dfrac{\partial f}{\partial y}$

(5) $\theta$ ... $x_0 \longrightarrow x_1 \longrightarrow x_2$, $f_1$, $f_2$  $\mathbb{E}_{x_1,x_2}\left[f_1(x_1)+f_2(x_2)\right]$  $\dfrac{\partial}{\partial \theta}\log p(x_1\mid\theta,x_0)(f_1(x_1)+f_2(x_2))$
$+\dfrac{\partial}{\partial \theta}\log p(x_2\mid\theta,x_1)f_2(x_2)$

Figure 1: Simple stochastic computation graphs

These simple examples illustrate several important motifs, where stochastic and deterministic nodes are arranged in series or in parallel. For example, note that in (2) the derivative of $y$ does not appear in the estimator, since the path from $\theta$ to $f$ is "blocked" by $x$. Similarly, in (3), $p(y\mid x)$ does not appear (this type of behavior is particularly useful if we only have access to a simulator of a system, but not access to the actual likelihood function). On the other hand, (4) has a direct path from $\theta$ to $f$, which contributes a term to the gradient estimator. (5) resembles a parameterized Markov reward process, and it illustrates that we'll obtain score function terms of the form *grad log-probability $\times$ future costs*.

The examples above all have one input $\theta$, but the formalism accommodates models with multiple inputs, for example a stochastic neural network with multiple layers of weights and biases, which may influence different subsets of the stochastic and cost nodes. See Appendix C for nontrivial examples with stochastic nodes and multiple inputs. The figure on the right shows a deterministic computation graph representing classification loss for a two-layer neural network, which has four parameters $(W_1, b_1, W_2, b_2)$ (weights and biases). Of course, this deterministic computation graph is a special type of stochastic computation graph.

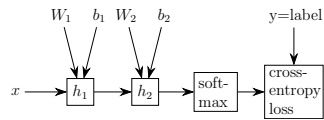

# 3 Main Results on Stochastic Computation Graphs

## 3.1 Gradient Estimators

This section will consider a general stochastic computation graph, in which a certain set of nodes are designated as *costs*, and we would like to compute the gradient of the sum of costs with respect to some input node $\theta$.

In brief, the main results of this section are as follows:

1. We derive a gradient estimator for an expected sum of costs in a stochastic computation graph. This estimator contains two parts (1) a *score function* part, which is a sum of terms *grad log-prob of variable × sum of costs influenced by variable*; and (2) a *pathwise derivative* term, that propagates the dependence through differentiable functions.

2. This gradient estimator can be computed efficiently by differentiating an appropriate "surrogate" objective function.

Let $\Theta$ denote the set of input nodes, $\mathcal{D}$ the set of deterministic nodes, and $\mathcal{S}$ the set of stochastic nodes. Further, we will designate a set of cost nodes $\mathcal{C}$, which are scalar-valued and deterministic. (Note that there is no loss of generality in assuming that the costs are deterministic—if a cost is stochastic, we can simply append a deterministic node that applies the identity function to it.) We will use $\theta$ to denote an input node ($\theta \in \Theta$) that we differentiate with respect to. In the context of machine learning, we will usually be most concerned with differentiating with respect to a parameter vector (or tensor), however, the theory we present does not make any assumptions about what $\theta$ represents.

For the results that follow, we need to define the notion of "influence", for which we will introduce two relations $\prec$ and $\prec^D$. The relation $v \prec w$ ("v influences w") means that there exists a sequence of nodes $a_1, a_2, \dots, a_K$, with $K \geq 0$, such that $(v, a_1), (a_1, a_2), \dots, (a_{K-1}, a_K), (a_K, w)$ are edges in the graph. The relation $v \prec^D w$ ("v *deterministically* influences w") is defined similarly, except that now we require that each $a_k$ is a deterministic node. For example, in Figure 1, diagram (5) above, $\theta$ influences $\{x_1, x_2, f_1, f_2\}$, but it only deterministically influences $\{x_1, x_2\}$.

Next, we will establish a condition that is sufficient for the existence of the gradient. Namely, we will stipulate that every edge $(v, w)$ with $w$ lying in the "influenced" set of $\theta$ corresponds to a differentiable dependency: if $w$ is deterministic, then the Jacobian $\frac{\partial w}{\partial v}$ must exist; if $w$ is stochastic, then the probability mass function $p(w \mid v, \dots)$ must be differentiable with respect to $v$.

> **Notation Glossary**
>
> $\Theta$: Input nodes
>
> $\mathcal{D}$: Deterministic nodes
>
> $\mathcal{S}$: Stochastic nodes
>
> $\mathcal{C}$: Cost nodes
>
> $v \prec w$: $v$ influences $w$
>
> $v \prec^D w$: $v$ deterministically influences $w$
>
> $\text{DEPS}_v$: "dependencies", $\{w \in \Theta \cup \mathcal{S} \mid w \prec^D v\}$
>
> $\hat{Q}_v$: sum of cost nodes influenced by $v$.
>
> $\hat{v}$: denotes the sampled value of the node $v$.

More formally:

> **Condition 1** (Differentiability Requirements). *Given input node $\theta \in \Theta$, for all edges $(v, w)$ which satisfy $\theta \prec^D v$ and $\theta \prec^D w$, then the following condition holds: if $w$ is deterministic, Jacobian $\frac{\partial w}{\partial v}$ exists, and if $w$ is stochastic, then the derivative of the probability mass function $\frac{\partial}{\partial v} p(w \mid \text{PARENTS}_w)$ exists.*

Note that Condition 1 does not require that all the functions in the graph are differentiable. If the path from an input $\theta$ to deterministic node $v$ is blocked by stochastic nodes, then $v$ may be a nondifferentiable function of its parents. If a path from input $\theta$ to stochastic node $v$ is blocked by other stochastic nodes, the likelihood of $v$ given its parents need not be differentiable; in fact, it does not need to be known[2].

We need a few more definitions to state the main theorems. Let $\text{DEPS}_v := \{w \in \Theta \cup \mathcal{S} \mid w \prec^D v\}$, the "dependencies" of node $v$, i.e., the set of nodes that deterministically influence it. Note the following:

- If $v \in \mathcal{S}$, the probability mass function of $v$ is a function of $\text{DEPS}_v$, i.e., we can write $p(v \mid \text{DEPS}_v)$.
- If $v \in \mathcal{D}$, $v$ is a deterministic function of $\text{DEPS}_v$, so we can write $v(\text{DEPS}_v)$.

Let $\hat{Q}_v := \sum_{\substack{c \succ v, \\ c \in \mathcal{C}}} \hat{c}$, i.e., the sum of costs downstream of node $v$. These costs will be treated as constant, fixed to the values obtained during sampling. In general, we will use the hat symbol $\hat{v}$ to denote a sample value of variable $v$, which will be treated as constant in the gradient formulae.

Now we can write down a general expression for the gradient of the expected sum of costs in a stochastic computation graph:

**Theorem 1.** *Suppose that $\theta \in \Theta$ satisfies Condition 1. Then the following two equivalent equations hold:*

$$\frac{\partial}{\partial \theta} \mathbb{E}\left[\sum_{c \in \mathcal{C}} c\right] = \mathbb{E}\left[\sum_{\substack{w \in \mathcal{S}, \\ \theta \prec^D w}} \left(\frac{\partial}{\partial \theta} \log p(w \mid \text{DEPS}_w)\right) \hat{Q}_w + \sum_{\substack{c \in \mathcal{C} \\ \theta \prec^D c}} \frac{\partial}{\partial \theta} c(\text{DEPS}_c)\right] \quad (5)$$

$$= \mathbb{E}\left[\sum_{c \in \mathcal{C}} \hat{c} \sum_{\substack{w \prec c, \\ \theta \prec^D w}} \frac{\partial}{\partial \theta} \log p(w \mid \text{DEPS}_w) + \sum_{\substack{c \in \mathcal{C}, \\ \theta \prec^D c}} \frac{\partial}{\partial \theta} c(\text{DEPS}_c)\right] \quad (6)$$

**Proof**: See Appendix A.

The estimator expressions above have two terms. The first term is due to the influence of $\theta$ on probability distributions. The second term is due to the influence of $\theta$ on the cost variables through a chain of differentiable functions. The distribution term involves a sum of gradients times "downstream" costs. The first term in Equation (5) involves a sum of gradients times "downstream" costs, whereas the first term in Equation (6) has a sum of costs times "upstream" gradients.

### 3.2 Surrogate Loss Functions

The next corollary lets us write down a "surrogate" objective $L$, which is a function of the inputs that we can differentiate to obtain an unbiased gradient estimator.

**Corollary 1.** *Let $L(\Theta, \mathcal{S}) := \sum_w \log p(w \mid \text{DEPS}_w) \hat{Q}_w + \sum_{c \in \mathcal{C}} c(\text{DEPS}_c)$. Then differentiation of $L$ gives us an unbiased gradient estimate: $\frac{\partial}{\partial \theta} \mathbb{E}\left[\sum_{c \in \mathcal{C}} c\right] = \mathbb{E}\left[\frac{\partial}{\partial \theta} L(\Theta, \mathcal{S})\right]$.*

One practical consequence of this result is that we can apply a standard automatic differentiation procedure to $L$ to obtain an unbiased gradient estimator. In other words, we convert the stochastic computation graph into a deterministic computation graph, to which we can apply the backpropagation algorithm.

There are several alternative ways to define the surrogate objective function that give the same gradient as $L$ from Corollary 1. We could also write $L(\Theta, \mathcal{S}) := \sum_w \frac{p(\hat{w} \mid \text{DEPS}_w)}{\hat{P}_v} \hat{Q}_w + \sum_{c \in \mathcal{C}} c(\text{DEPS}_c)$, where $\hat{P}_w$ is the probability $p(w \mid \text{DEPS}_w)$ obtained during sampling, which is viewed as a constant.

The surrogate objective from Corollary 1 is actually an upper bound on the true objective in the case that (1) all costs $c \in \mathcal{C}$ are negative,

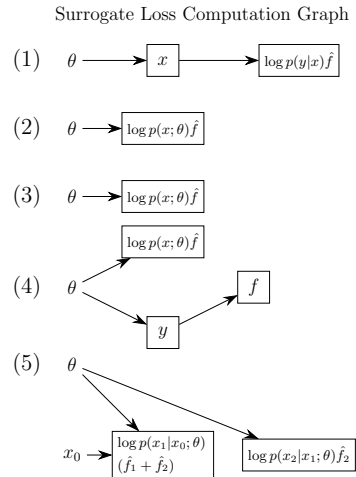

Surrogate Loss Computation Graph

Figure 2: Deterministic computation graphs obtained as surrogate loss functions of stochastic computation graphs from Figure 1.

(2) the the costs are not deterministically influenced by the parameters $\Theta$. This construction allows from majorization-minimization algorithms (similar to EM) to be applied to general stochastic computation graphs. See Appendix B for details.

### 3.3 Higher-Order Derivatives.

The gradient estimator for a stochastic computation graph is itself a stochastic computation graph. Hence, it is possible to compute the gradient yet again (for each component of the gradient vector), and get an estimator of the Hessian. For most problems of interest, it is not efficient to compute this dense Hessian. On the other hand, one can also differentiate the gradient-vector product to get a Hessian-vector product—this computation is usually not much more expensive than the gradient computation itself. The Hessian-vector product can be used to implement a quasi-Newton algorithm via the conjugate gradient algorithm [28]. A variant of this technique, called Hessian-free optimization [13], has been used to train large neural networks.

## 4 Variance Reduction

Consider estimating $\frac{\partial}{\partial \theta} \mathbb{E}_{x \sim p(\cdot;\, \theta)} [f(x)]$. Clearly this expectation is unaffected by subtracting a constant $b$ from the integrand, which gives $\frac{\partial}{\partial \theta} \mathbb{E}_{x \sim p(\cdot;\, \theta)} [f(x) - b]$. Taking the score function estimator, we get $\frac{\partial}{\partial \theta} \mathbb{E}_{x \sim p(\cdot;\, \theta)} [f(x)] = \mathbb{E}_{x \sim p(\cdot;\, \theta)} \left[ \frac{\partial}{\partial \theta} \log p(x;\, \theta)(f(x) - b) \right]$. Taking $b = \mathbb{E}_x [f(x)]$ generally leads to substantial variance reduction—$b$ is often called a *baseline*[3] (see [6] for a more thorough discussion of baselines and their variance reduction properties).

We can make a general statement for the case of stochastic computation graphs—that we can add a baseline to every stochastic node, which depends all of the nodes it doesn't influence. Let NONINFLUENCED$(v) := \{w \mid v \nprec w\}$.

**Theorem 2.**

$$
\frac{\partial}{\partial \theta} \mathbb{E} \left[ \sum_{c \in \mathcal{C}} c \right] = \mathbb{E} \left[ \sum_{\substack{v \in \mathcal{S} \\ v \succ \theta}} \left( \frac{\partial}{\partial \theta} \log p(v \mid \text{PARENTS}_v) \right) (\hat{Q}_v - b(\text{NONINFLUENCED}(v)) + \sum_{c \in \mathcal{C} \succeq \theta} \frac{\partial}{\partial \theta} c \right]
$$

**Proof**: See Appendix A.

## 5 Algorithms

As shown in Section 3, the gradient estimator can be obtained by differentiating a surrogate objective function $L$. Hence, this derivative can be computed by performing the backpropagation algorithm on $L$. That is likely to be the most practical and efficient method, and can be facilitated by automatic differentiation software.

Algorithm 1 shows explicitly how to compute the gradient estimator in a backwards pass through the stochastic computation graph. The algorithm will recursively compute $g_v := \frac{\partial}{\partial v} \mathbb{E} \left[ \sum_{\substack{c \in \mathcal{C} \\ v \prec c}} c \right]$ at every deterministic and input node $v$.

## 6 Related Work

As discussed in Section 2, the score function and pathwise derivative estimators have been used in a variety of different fields, under different names. See [3] for a review of gradient estimation, mostly from the simulation optimization literature. Glasserman's textbook provides an extensive treatment of various gradient estimators and Monte Carlo estimators in general. Griewank and Walther's textbook [8] is a comprehensive reference on computation graphs and automatic differentiation (of deterministic programs.) The notation and nomenclature we use is inspired by Bayes nets and influence diagrams [19]. (In fact, a stochastic computation graph is a type of Bayes network; where the deterministic nodes correspond to degenerate probability distributions.)

The topic of gradient estimation has drawn significant recent interest in machine learning. Gradients for networks with stochastic units was investigated in Bengio et al. [2], though they are concerned

**Algorithm 1** Compute Gradient Estimator for Stochastic Computation Graph

---
    **for** $v \in$ Graph **do**                                                                 ▷ Initialization at output nodes

$$\mathbf{g}_v = \begin{cases} \mathbf{1}_{\dim v} & \text{if } v \in \mathcal{C} \\ \mathbf{0}_{\dim v} & \text{otherwise} \end{cases}$$

    **end for**
    Compute $\hat{Q}_w$ for all nodes $w \in$ Graph
    **for** $v$ in REVERSETOPOLOGICALSORT(NONINPUTS) **do**                      ▷ Reverse traversal
        **for** $w \in$ PARENTS$_v$ **do**
            **if not** ISSTOCHASTIC($w$) **then**
                **if** ISSTOCHASTIC($v$) **then**
                    $\mathbf{g}_w \mathrel{+}= (\frac{\partial}{\partial w} \log p(v \mid \text{PARENTS}_v))\hat{Q}_w$
                **else**
                    $\mathbf{g}_w \mathrel{+}= (\frac{\partial v}{\partial w})^T \mathbf{g}_v$
                **end if**
            **end if**
        **end for**
    **end for**
    **return** $[\mathbf{g}_\theta]_{\theta \in \Theta}$

---

with differentiating through individual units and layers; not how to deal with arbitrarily structured models and loss functions. Kingma and Welling [11] consider a similar framework, although only with continuous latent variables, and point out that reparameterization can be used to to convert hierarchical Bayesian models into neural networks, which can then be trained by backpropagation.

The score function method is used to perform variational inference in general models (in the context of probabilistic programming) in Wingate and Weber [27], and similarly in Ranganath et al. [20]; both papers mostly focus on mean-field approximations without amortized inference. It is used to train generative models using neural networks with discrete stochastic units in Mnih and Gregor [14] and Gregor et al. in [7]; both amortize inference by using an inference network.

Generative models with continuous valued latent variables networks are trained (again using an inference network) with the reparametrization method by Rezende, Mohamed, and Wierstra [21] and by Kingma and Welling [10]. Rezende et al. also provide a detailed discussion of reparameterization, including a discussion comparing the variance of the SF and PD estimators.

Bengio, Leonard, and Courville [2] have recently written a paper about gradient estimation in neural networks with stochastic units or non-differentiable activation functions—including Monte Carlo estimators and heuristic approximations. The notion that policy gradients can be computed in multiple ways was pointed out in early work on policy gradients by Williams [26]. However, all of this prior work deals with specific structures of the stochastic computation graph and does not address the general case.

# 7 Conclusion

We have developed a framework for describing a computation with stochastic and deterministic operations, called a stochastic computation graph. Given a stochastic computation graph, we can automatically obtain a gradient estimator, given that the graph satisfies the appropriate conditions on differentiability of the functions at its nodes. The gradient can be computed efficiently in a backwards traversal through the graph: one approach is to apply the standard backpropagation algorithm to one of the surrogate loss functions from Section 3; another approach (which is roughly equivalent) is to apply a modified backpropagation procedure shown in Algorithm 1. The results we have presented are sufficiently general to automatically reproduce a variety of gradient estimators that have been derived in prior work in reinforcement learning and probabilistic modeling, as we show in Appendix C. We hope that this work will facilitate further development of interesting and expressive models.

# 8 Acknowledgements

We would like to thank Shakir Mohamed, Dave Silver, Yuval Tassa, Andriy Mnih, and others at DeepMind for insightful comments.

## Footnotes

[1] Note that for the pathwise derivative estimator, $f(x(z, \theta))$ merely needs to be a *continuous* function of $\theta$—it is sufficient that this function is almost-everywhere differentiable. A similar statement can be made about $p(x; \theta)$ and the score function estimator. See Glasserman [4] for a detailed discussion of the technical requirements for these gradient estimators to be valid.

[2]This fact is particularly important for reinforcement learning, allowing us to compute policy gradient estimates despite having a discontinuous dynamics function or reward function.

[3]The optimal baseline for scalar $\theta$ is in fact the weighted expectation $\frac{\mathbb{E}_x [f(x)s(x)^2]}{\mathbb{E}_x [s(x)^2]}$ where $s(x) = \frac{\partial}{\partial \theta} \log p(x;\, \theta)$.

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
