[Supplementary Material]

## A  Proofs

### Theorem 1

We will consider the case that all of the random variables are continuous-valued, thus the expectations can be written as integrals. For discrete random variables, the integrals should be changed to sums.

Recall that we seek to compute $\frac{\partial}{\partial \theta} \mathbb{E} \left[ \sum_{c \in \mathcal{C}} c \right]$. We will differentiate the expectation of a single cost term; summing over these terms yields Equation (6).

$$\mathbb{E}_{\substack{v \in \mathcal{S}, \\ v \prec c}} [c] = \int \prod_{\substack{v \in \mathcal{S}, \\ v \prec c}} p(v \mid \text{DEPS}_v) dv \ c(\text{DEPS}_c) \tag{7}$$

$$\frac{\partial}{\partial \theta} \mathbb{E}_{\substack{v \in \mathcal{S}, \\ v \prec c}} [c] = \frac{\partial}{\partial \theta} \int \prod_{\substack{v \in \mathcal{S}, \\ v \prec c}} p(v \mid \text{DEPS}_v) dv \ c(\text{DEPS}_c) \tag{8}$$

$$= \int \prod_{\substack{v \in \mathcal{S}, \\ v \prec c}} p(v \mid \text{DEPS}_v) dv \sum_{\substack{v \in \mathcal{S}, \\ v \prec c}} \left[ \frac{\frac{\partial}{\partial \theta} p(v \mid \text{DEPS}_v)}{p(v \mid \text{DEPS}_v)} c(\text{DEPS}_c) + \frac{\partial}{\partial \theta} c(\text{DEPS}_c) \right] \tag{9}$$

$$= \int \prod_{\substack{v \in \mathcal{S}, \\ v \prec c}} p(v \mid \text{DEPS}_v) dv \sum_{\substack{v \in \mathcal{S}, \\ v \prec c}} \left[ \left( \frac{\partial}{\partial \theta} \log p(v \mid \text{DEPS}_v) \right) c(\text{DEPS}_c) + \frac{\partial}{\partial \theta} c(\text{DEPS}_c) \right] \tag{10}$$

$$= \mathbb{E}_{\substack{v \in \mathcal{S}, \\ v \prec c}} \left[ \frac{\partial}{\partial \theta} \log p(v \mid \text{DEPS}_v) \hat{c} + \frac{\partial}{\partial \theta} c(\text{DEPS}_c) \right] \tag{11}$$

Equation (9) requires that the integrand is differentiable, which is satisfied if all of the PDFs and $c(\text{DEPS}_c)$ are differentiable. Equation (6) follows by summing over all costs $c \in \mathcal{C}$. Equation (5) follows from rearrangement of the terms in this equation.

### Theorem 2

It suffices to show that for a particular node $v \in \mathcal{S}$, the following expectation (taken over all variables) vanishes

$$\mathbb{E} \left[ \left( \frac{\partial}{\partial \theta} \log p(v \mid \text{PARENTS}_v) \right) b(\text{NONINFLUENCED}(v)) \right] . \tag{12}$$

Analogously to $\text{NONINFLUENCED}(v)$, define $\text{INFLUENCED}(v) := \{w \mid w \succ v\}$. Note that the nodes can be ordered so that $\text{NONINFLUENCED}(v)$ all come before $v$ in the ordering. Thus, we can write

$$\mathbb{E}_{\text{NONINFLUENCED}(v)} \left[ \mathbb{E}_{\text{INFLUENCED}(v)} \left[ \left( \frac{\partial}{\partial \theta} \log p(v \mid \text{PARENTS}_v) \right) b(\text{NONINFLUENCED}(v)) \right] \right] \tag{13}$$

$$= \mathbb{E}_{\text{NONINFLUENCED}(v)} \left[ \mathbb{E}_{\text{INFLUENCED}(v)} \left[ \left( \frac{\partial}{\partial \theta} \log p(v \mid \text{PARENTS}_v) \right) \right] b(\text{NONINFLUENCED}(v)) \right] \tag{14}$$

$$= \mathbb{E}_{\text{NONINFLUENCED}(v)} \left[ 0 \cdot b(\text{NONINFLUENCED}(v)) \right] \tag{15}$$

$$= 0 \tag{16}$$

where we used $\mathbb{E}_{\text{INFLUENCED}(v)} \left[ \left( \frac{\partial}{\partial \theta} \log p(v \mid \text{PARENTS}_v) \right) \right] = \mathbb{E}_v \left[ \left( \frac{\partial}{\partial \theta} \log p(v \mid \text{PARENTS}_v) \right) \right] = 0$.

## B  Surrogate as an Upper Bound, and MM Algorithms

$L$ has additional significance besides allowing us to estimate the gradient of the expected sum of costs. Under certain conditions, $L$ is a upper bound on on the true objective (plus a constant).

We shall make two restrictions on the stochastic computation graph: (1) first, that all costs $c \in \mathcal{C}$ are negative. (2) the the costs are not deterministically influenced by the parameters $\Theta$. First, let

us use importance sampling to write down the expectation of a given cost node, when the sampling distribution is different from the distribution we are evaluating: for parameter $\theta \in \Theta$, $\theta = \theta_{\text{old}}$ is used for sampling, but we are evaluating at $\theta = \theta_{\text{new}}$.

$$
\mathbb{E}_{v \prec c \mid \theta_{\text{new}}} [\hat{c}] = \mathbb{E}_{v \prec c \mid \theta_{\text{old}}} \left[ \hat{c} \prod_{\substack{v \prec c, \\ \theta \prec^D v}} \frac{P_v(v \mid \text{DEPS}_v \backslash \theta, \theta_{\text{new}})}{P_v(v \mid \text{DEPS}_v \backslash \theta, \theta_{\text{old}})} \right] \tag{17}
$$

$$
\leq \mathbb{E}_{v \prec c \mid \theta_{\text{old}}} \left[ \hat{c} \left( \log \left( \prod_{\substack{v \prec c, \\ \theta \prec^D v}} \frac{P_v(v \mid \text{DEPS}_v \backslash \theta, \theta_{\text{new}})}{P_v(v \mid \text{DEPS}_v \backslash \theta, \theta_{\text{old}})} \right) + 1 \right) \right] \tag{18}
$$

where the second line used the inequality $x \geq \log x + 1$, and the sign is reversed since $\hat{c}$ is negative. Summing over $c \in \mathcal{C}$ and rearranging we get

$$
\mathbb{E}_{\mathcal{S} \mid \theta_{\text{new}}} \left[ \sum_{c \in \mathcal{C}} \hat{c} \right] \leq \mathbb{E}_{\mathcal{S} \mid \theta_{\text{old}}} \left[ \sum_{c \in \mathcal{C}} \hat{c} + \sum_{v \in \mathcal{S}} \log \left( \frac{p(v \mid \text{DEPS}_v \backslash \theta, \theta_{\text{new}})}{p(v \mid \text{DEPS}_v \backslash \theta, \theta_{\text{old}})} \right) \hat{Q}_v \right] \tag{19}
$$

$$
= \mathbb{E}_{\mathcal{S} \mid \theta_{\text{old}}} \left[ \sum_{v \in \mathcal{S}} \log p(v \mid \text{DEPS}_v \backslash \theta, \theta_{\text{new}}) \hat{Q}_v \right] + \text{const} \tag{20}
$$

Equation (20) allows for majorization-minimization algorithms (like the EM algorithm) to be used to optimize with respect to $\theta$. In fact, similar equations have been derived by interpreting rewards (negative costs) as probabilities, and then taking the variational lower bound on log-probability (e.g., [24]).

## C Examples

### C.1 Generalized EM Algorithm and Variational Inference.

The generalized EM algorithm maximizes likelihood in a probabilistic model with latent variables [18]. Suppose the probabilistic model defines a probability distribution $p(x, z; \theta)$ where $x$ is observed, $z$ is a latent variable, and $\theta$ is a parameter of the distribution. The generalized EM algorithm maximizes the *variational lower bound*, which is defined by an expectation over $q$:

$$
L(\theta, q) = \mathbb{E}_{z \sim q} \left[ \log \left( \frac{p(x, z; \theta)}{q(z)} \right) \right]. \tag{21}
$$

The generalized EM algorithm can take many different forms, leading to different gradient estimation problems.

**Neural variational inference.** [14] propose a generalized EM algorithm for multi-layered latent variable models such as sigmoidal belief networks that employs an *inference network*, an explicit parameterization of $q$ as a function of the observed data $x$, to allow for fast approximate inference. The generative model and inference network take the form

$$
p_\theta(x) = \sum_{h_1, h_2} p_{\theta_1}(x|h_1) p_{\theta_2}(h_1|h_2) p_{\theta_3}(h_2|h_3) p_{\theta_3}(h_3)
$$

$$
q_\phi(h_1, h_2|x) = q_{\phi_1}(h_1|x) q_{\phi_2}(h_2|h_1) q_{\phi_3}(h_3|h_2),
$$

and thus

$$
L(\theta, \phi) = \mathbb{E}_{h \sim q_\phi} \left[ \underbrace{\log \frac{p_{\theta_1}(x|h_1)}{q_{\phi_1}(h_1|x)}}_{=r_1} + \underbrace{\log \frac{p_{\theta_2}(h_1|h_2)}{q_{\phi_2}(h_2|h_1)}}_{=r_2} + \underbrace{\log \frac{p_{\theta_3}(h_2|h_3) p_{\theta_3}(h_3)}{q_{\phi_3}(h_3|h_2)}}_{=r_3} \right].
$$

Given a sample $h \sim q_\phi$ an unbiased estimate of the gradient is obtained

$$\frac{\partial L}{\partial \theta} \approx \frac{\partial}{\partial \theta} \log p_{\theta_1}(x|h_1) + \frac{\partial}{\partial \theta} \log p_{\theta_2}(h_1|h_2) + \frac{\partial}{\partial \theta} \log p_{\theta_3}(h_2) \tag{22}$$

$$\frac{\partial L}{\partial \phi} \approx \frac{\partial}{\partial \phi} \log q_{\phi_1}(h_1|x)(\hat{Q}_1 - b_1(x)) + \frac{\partial}{\partial \phi} \log q_{\phi_2}(h_2|h_1)(\hat{Q}_2 - b_2(h_1)) + \frac{\partial}{\partial \phi} \log q_{\phi_3}(h_3|h_2)(\hat{Q}_3 - b_3(h_2)) \tag{23}$$

where $\hat{Q}_1 = r_1 + r_2 + r_3$; $\hat{Q}_2 = r_2 + r_3$; and $\hat{Q}_3 = r_3$. Eq. (22) uses the PD estimator to estimate the gradient with respect to the model parameters $\theta$; eq. (23) is an application of the SF estimator to the gradient with respect to the parameters $\phi$ of the inference network; $b_1, b_2, b_3$ are baselines.

**Variational Autoencoder, Deep Latent Gaussian Models and Reparameterization.** [10, 21] consider a similar formulation to [14] but have continuous latent variables and can thus re-parameterize their inference network to enable the use of the PD estimator:

$$L_{\text{orig}}(\theta, \phi) = \mathbb{E}_{h \sim q_\phi}\left[\log \frac{p_\theta(x|h)p_\theta(h)}{q_\phi(h|x)}\right] \tag{24}$$

$$L_{\text{re}}(\theta, \phi) = \mathbb{E}_{\epsilon \sim \rho}\left[\log p_\theta(x|h_\phi(\epsilon, x)) + \log p_\theta(h_\phi(\epsilon, x))\right] + H[q_\phi(\cdot|x)] \tag{25}$$

where the second term, the entropy of $q_\phi$ can be computed analytically for the parametric forms of $q$ considered in the paper (Gaussians). For $q_\phi$ being conditionally Gaussian, i.e. $q_\phi(h|x) = N(h|\mu_\phi(x), \sigma_\phi(x))$ reparameterizing leads e.g. to $h = h_\phi(\epsilon; x) = \mu_\phi(x) + \epsilon \sigma_\phi(x)$.

Reparameterization

Given $\epsilon \sim \rho$ an estimate of the gradient is obtained as

$$\frac{\partial L_{\text{re}}}{\partial \theta} \approx \frac{\partial}{\partial \theta}\left[\log p_\theta(x|h_\phi(\epsilon, x)) + \log p_\theta(h_\phi(\epsilon, x))\right], \tag{26}$$

$$\frac{\partial L_{\text{re}}}{\partial \phi} \approx \left[\frac{\partial}{\partial h} \log p_\theta(x|h_\phi(\epsilon, x)) + \frac{\partial}{\partial h} \log p_\theta(h_\phi(\epsilon, x))\right] \frac{\partial h}{\partial \phi} + \frac{\partial}{\partial \phi} H[q_\phi(\cdot|x)] \tag{27}$$

### C.2 Policy Gradients in Reinforcement Learning.

In reinforcement learning, an agent interacts with an environment according to its policy $\pi$ and receives a reward. The goal is to maximize the expected sum of rewards, the return, under the trajectory distribution that is specified jointly by the environment dynamics and the policy. Policy gradient methods seek to directly estimate the gradient of expected return with respect to the policy parameters [26, 1, 23]. The RL case is especially interesting as we typically assume that the environment dynamics are not available analytically and can only be sampled, which has implication for gradient estimation. Below we distinguish two important cases: *Markov decision processes* (MDP) and *partially observable Markov decision processes* (POMDP).

**MDPs**: In the MDP case, the expectation is taken with respect to the distribution over state ($s$) and action ($a$) sequences

$$L(\theta) = \mathbb{E}_{\tau \sim p_\theta}\left[\sum_{t=1}^{T} r(s_t, a_t)\right], \tag{28}$$

where $\tau = (s_1, a_1, s_2, a_2, \dots)$ are trajectories and the distribution over trajectories is defined in terms of the environment dynamics $p_E(s_{t+1}|s_t, a_t)$ and the policy $\pi_\theta$: $p_\theta(\tau) = p_E(s_1) \prod_t \pi_\theta(a_t|s_t) p_E(s_{t+1}|s_t, a_t)$. $r$ are rewards (negative costs in the terminology of the rest of

the paper). The classic *REINFORCE* [26] estimate of the gradient is given by

$$\frac{\partial}{\partial \theta} L = \mathbb{E}_{\tau \sim p_\theta} \left[ \sum_{t=1}^{T} \frac{\partial}{\partial \theta} \log \pi_\theta(a_t|s_t) \left( \sum_{t'=t}^{T} r(s_{t'}, a_{t'}) - b_t(s_t) \right) \right], \tag{29}$$

where $b_t(s_t)$ is an arbitrary baseline which is often chosen to be $V_t(s_t) = \mathbb{E}_{\tau \sim p_\theta} \left[ \sum_{t'=t}^{T} r(s_{t'}, a_{t'}) \right]$, i.e. the well-known state-value function. (Equation (29)) corresponds to an application of the SF estimator at the stochastic nodes $a_t$. It is worth noting that a Monte Carlo estimate of (Equation (29)) only requires simulating from the environment by running trajectories forward according to the current policy. This is due to the property of the SF estimator which only requires evaluation (sampling in the stochastic case) of the nodes downstream of the stochastic node $a_t$

**POMDPs.**

POMDPs differ from MDPs in that the state $s_t$ of the environment is not observed directly but, as in latent-variable time series models, only through stochastic observations $o_t$, which depend on the latent states $s_t$ via $p_E(o_t|s_t)$. The policy therefore has to be a function of the history of past observations $\pi_\theta(a_t|o_1 \ldots o_t)$. For instance it can take the form of a recurrent neural network (RNN) [25, 15]. A *REINFORCE* gradient estimate is then given by

$$\frac{\partial}{\partial \theta} L = \mathbb{E}_{\tau \sim p_\theta} \Big[ \sum_{t=1}^{T} \frac{\partial}{\partial \theta} \log \pi_\theta(a_t|o_1 \ldots o_t))$$

$$\left( \sum_{t'=t}^{T} r(s_{t'}, a_{t'}) - b_t(o_1 \ldots o_t) \right) \Big]. \tag{30}$$

Note that, at each time step $t$, the gradient $\frac{\partial}{\partial \theta} \log \pi_\theta$ at the stochastic node $a_t$ is estimated using the SF estimator, and then backpropagated in the RNN via chain-rule in the usual manner. As before, $b_t$ is a baseline, which is written here as a function of the observation history up to time $t$ and, as the policy, which can be parameterized through another RNN.