[Reviews · NeurIPS 2015]

Submitted by Assigned_Reviewer_1

This paper introduces stochastic computation graphs as a framework in which to consider both score-function estimators

of stochastic gradients, and estimators based on reparamterization of continuous distributions. Once a computation is represented as a stochastic computation graph, a deterministic surrogate function (which can be automatically differentiated) can be defined. The pathwise derivative estimator is used when possible, otherwise the score function estimator.

The main value of this paper is to make explicit a manner for constructing estimators for these stochastic gradients, in a way which is incredibly clearly presented, and unifies work from variational inference, reinforcement learning, neural network communities. On some level, the amount of novel content is low - the originality lies in the presentation.

The two main theorems (general representation, and variance reduction through control variates) are in some sense the results one would expect. However, I believe this representation is quite useful - one could imagine, now, an automated tool for

constructing efficient stochastic gradient estimators given program source code that contains random choices,

in a manner similar to how tools such as Theano can construct gradients for deterministic functions automatically. A set of straightforward, understandable results for computing stochastic gradients in a general setting should have wide applicability within machine learning.

There is a typo in appendix A, 510-511 ("Clearly ...").
Summary: This is enjoyable paper to read. It provides a nice discussion of approaches to stochastic gradient estimations which have both seen much recent interest, and introduces a formalism in which to discuss them.

Submitted by Assigned_Reviewer_2

The authors propose to unify a variety of gradient-based learning methods that involve stochastic and deterministic computations under a formalism called stochastic computation graphs. In this setting, they show that the gradient of the expected loss can be efficiently computed through backpropagation of the gradient for a surrogate loss obtained by introducing extra gradient terms for the stochastic nodes. Additionally, they explain how to do Hessian-vector products and describe some variance reduction techniques. The paper is very well written and the authors went to great lengths to introduce and explain the concepts clearly. Some minor typos should be corrected: - line 245: remove the word "is" - line 260: replace x with theta - line 270: replace V with v (also in the Notation Glossary)
Summary: Good paper, clearly written, that generalizes a variety of gradient-based learning methods using stochastic computation graphs. The paper can be accepted after some minor corrections.

Submitted by Assigned_Reviewer_3

The manuscript proposes a formalism for computing stochastic estimates of gradients for loss functions.

The formalism, referred to as stochastic computation graphs, is very general, applying to models with deterministic and stochastic components, and allowing the computation of gradient estimates for a broad range of models.

Methods for deep networks, variational inference, and reinforcement learning are identified as special cases of the proposed framework.

The proposed stochastic computation graphs are essentially Bayesian networks which may also contain deterministic nodes, and which are interpreted as encoding distributions over loss functions that are to be minimized.

The gradient estimation algorithm extends backpropagation to the partially stochastic case covered by these models, by simply composing score function estimators and pathwise derivative estimators.

The main result of the paper is a theorem stating that the gradient estimation algorithm recovers an unbiased estimate of the gradient for these models, under broad conditions.

The formalism and algorithm are both simple and represent fairly small adaptions of existing ideas.

However, the ideas are likely to have substantial practical impact by facilitating automatic optimization of a wide range of custom models and architectures.

The approach is both elegant and general.

The biggest weakness of the paper is a complete lack of empirical analysis.

This cannot be excused by considering it a "theory paper," since the proposed algorithm is practically motivated, and derives much of its potential impact from its potential use.

It would GREATLY strengthen the paper to include experimental results demonstrating that the method is competitive with state of the art special-purpose methods on a variety of real world problems.

It would also be insightful to explore and graph its performance on randomly generated problems, varying difficulty parameters such as the depth of the networks, percent of stochastic nodes, variance of stochastic nodes, number of parameters, etc.

The only mitigating factor regarding the lack of empirical results is that special cases of the method are identified in the literature, thereby indirectly demonstrating that it can have practical utility.

Together with the potential impact of the work, this leads me to consider it for publication, despite the above reservations.
Summary: The authors introduce a graphical formalism and an associated algorithm for computing stochastic gradient estimates of loss functions in models with both stochastic and deterministic components.

This is exciting work with potentially substantial impact for both deep learning and hierarchical probabilistic modeling.

The main weakness of the paper is an absence of experimental validation, although this is somewhat mitigated by the fact that a number of published (and therefore previously validated) methods are special cases of the proposed method.

Author Feedback
Author rebuttal: Thanks for all your feedback and suggestions. We will carefully incorporate them into our paper.

As observed by the reviewers, the contribution of the paper is theoretical: we derive formulas and algorithms for computing unbiased gradient estimators, after introducing a general formalism that provides a unifying view of a range of problems from reinforcement learning to probabilistic modeling. As noted by several reviewers, the methods derived in this paper are a generalization of special-purpose methods proposed in prior specific problem instances, e.g. [14, 15, 24, 25], where state-of-the-art empirical results are obtained. We agree with reviewer #3 that our framework suggests several interesting empirical questions, but they are beyond the scope of this paper, as they would require a much deeper dive into a specific problem domain.